# Sustainable Provision of School Buildings in The Netherlands: An Empirical Productivity Analysis of Local Government School Building Operations

**Jos L. T. Blank** 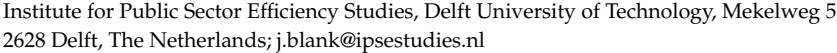

Institute for Public Sector Efficiency Studies, Delft University of Technology, Mekelweg 5, 2628 Delft, The Netherlands; j.blank@ipsestudies.nl

**Abstract:** Building operations and construction are responsible for a large part of global energy use and carbon dioxide emissions. In this paper, we present an analysis of the efficiency and productivity of the provision of school buildings by Dutch municipalities. A cost function is estimated for the years 2005–2016 using stochastic frontier methods based on data of Dutch municipalities. The results indicate that inefficiency and unproductiveness are substantial. The provision of school buildings on a more appropriate scale, detailed performance benchmarking and including more incentives for innovative behaviour may result in a more sustainable provision of school buildings and less energy use and emission of carbon dioxide.

**Keywords:** sustainability; efficiency; school buildings; productivity; local government; performance benchmarking; carbon dioxide

## 1. Introduction

Building operations and construction are responsible for 36% of global energy use and 39% of energy-related carbon dioxide emissions annually [1]. Emissions related to buildings come from two sources. The first source is the energy used during normal operations, such as lightning and heating, the so-called operational carbon emissions. The second source is the amount of carbon produced during construction, manufacturing building supplies and the transportation of materials to construction sites. The second source accounts for about 25% of a building's carbon emissions during its lifespan. Globally, the embodied carbon of buildings account for about 11% of emissions [1].

Obviously, buildings' carbon emissions are directly related to the construction process of the building, to the use of carbon-free or recyclable materials and to the extent to which the building is energy-efficient. However, another option is to promote the efficient usage of space in general. One of the most effective ways to reduce emissions is by diminishing the wastage of building capacity. Note that aside from all other technical and ecological improvements, measures to downsize overcapacity are free. One issue that is particular related to the capacity usage is the scale on which the firm or institution governing the buildings operates. A small firm has less flexibility in allocating building spaces and is less able to respond to changes in building capacity than a large firm. On the other hand, large firms may face bureaucratic issues and are less able to respond to changing capacities. One may think of information lacks and lengthy administrative procedures. In general, the scale issue may apply to all the infrastructural projects governments are dealing with, such as hospital, school, office and recreational buildings, but also to roads, canals, ports, etc. An interesting issue to address is which government layer should govern the provision of that specific infrastructure in order to downsize emissions.

In The Netherlands, more than 1.5 million primary education pupils are taught each day in 8500 school buildings. The school buildings cover an area exceeding 10 million

square metres (Court of Audit, 2016). In 2014, an estimated EUR 1.45 billion was spent on primary school buildings.

As a result of decentralisation, municipalities have been largely responsible for school buildings in both primary and secondary education since 1997. Until 1997, the central government was responsible for providing school buildings. The central idea behind the decentralisation was that local government would have a better insight into the needs and availability of school buildings. A more productive usage of existing capacities could be expected. Since primary and secondary schools are of a limited size, transferring responsibilities to school boards would have been a bridge too far. Aside from the capacity usage, some other economic issues may prevail as well. School board management probably lacks the specialised knowledge and insights in operating in the market of construction and real estate. It also is to be expected that their bargaining power is smaller than of local government. Furthermore, the financial risks and uncertainties for each separate school board would be too high. Therefore, the economic rationale behind the decentralisation was that economies of scale would prevail at the level of local government. In 2016, school building capacity was provided by about 400 municipalities, varying in population between 5000 and 820,000 inhabitants. Local government in The Netherlands has a large discretion power in how to spend their money. As long as they face no long-term budget deficits, central government cannot interfere with local policies. Political control is conducted by the inhabitants, who can vote for the local government board every four years. Whereas political power varies strongly amongst municipalities, substantial differences in local policies exist. This implies that municipality data are eminently suitable for research.

It is, therefore, an interesting question whether economies of scale, cost efficiencies and technical change prevail in the provision of school buildings by municipalities. Due to the fact that the size of local government varies so much amongst municipalities in The Netherlands, interesting data are available to test this hypothesis. The central research questions therefore are as follows:

1.　Do economies of scale exist in the provision of school buildings and is there an optimal scale?
2.　To what extent can each local government increase its cost efficiency?
3.　To what extent have local governments succeeded in improving school building productivity by innovative behaviour?

In this paper, we relate to these research questions by conducting an analysis of economies of scale, technical efficiency and technical change of providing school buildings by local government within the Dutch primary education system between 2005 and 2016.

In the next section, a cost model is estimated using stochastic frontier methods, using data over the period 2005–2016. The estimated frontier identifies the minimum cost (or volume) of school buildings given some level of output and contextual variables faced by the municipalities. From the results of the estimated cost function, economies of scale can be derived as well as (cost) efficiencies and technical change. In order to connect the methodology to the research questions, two issues must be emphasised. First, although we speak about cost function, we actually use a measure for the physical (building) input as the dependent variable. This measure is derived from cost by controlling for price effects (construction prices, interest rates and depreciation). Secondly, we measure services produced by the number of pupils actual using the buildings instead of the potential capacity (square or cubic meters, number of classrooms). This perspective allows us to derive conclusions from the outcomes in terms of productive usage of buildings or occupation rates.

For obvious reasons, the occupation rate is not the sole determinant of economies of scale or efficiency. Contracts on service and maintenance and administrative procedures may well affect these outcomes. Since we are lacking this type of information, we are not able to take these determinants into account and, therefore, some omitted variables bias may occur. However, we may assume that the impact of these determinants is relatively small compared to the impact of the occupation rate.

The remainder of this paper is organised as follows. The following section discusses the relevant literature. Section 3 outlines the methodology. Section 4 discusses the data, and Section 5 presents the estimation results. Section 6 concludes this paper.

## 2. Literature Review

In the literature, there is a strong focus on the efficiency of building operations, such as heating and lightning. These studies researched the effects of various determinants on building energy consumptions and described the relationship between environmental and managerial factors on energy use empirically [2]. Environmental factors are, for example, the outdoor temperature and building vintage [3,4]. Examples of managerial factors are indoor temperatures [5], the applications of artificial intelligent systems based on thermostats and sensors, collective holidays and additional isolation measures. In a number of studies, some measure of occupancy rate is also being included [5–7].

There is an extensive strand of the literature on the measurement and analysis of local government efficiency. Most of these studies focus on a specific service, such as waste collection [8–11], public health [12], policing [13–15], public administration [16,17], and public transport [18]. These studies also cover an extensive list of all kinds of themes, varying from scale issues, technical change to ways of tendering.

In a recent study, Niaounakis [19] particularly focuses on different issues regarding economies of scale in public services provided by local government. He analyses economies of scale in education, local infrastructures and tax offices from different perspectives. In his work, he stresses the relevance of the different scale concepts, relating to the size of the municipalities, to the size of the services, different governing layers and the public/private relationships. From this research two relevant conclusions emerge. First, the optimal size of public provision depends on the service. There is no such thing as one size fits all. Secondly, municipalities can benefit from scale economies by collaborating. Municipalities may vary the size of the collaborations over various services. Municipalities may optimally benefit from scale economies by letting the size of a collaboration depend on the service.

In spite of the numerous papers on municipality scale economies and efficiency, no research focusing on efficiency, scale economies or technical in the provision of school buildings has been produced. The only services that come close to it is the provision of rental family houses by corporations. They provide similar services, since they have to plan, design, construct, demolish, maintain and control properties. Interesting research concerns Dutch housing corporations [20,21]. He shows that the range for the optimal scale varies between 501 and 1000 dwellings. Corporations with more than 2500 dwellings strong diseconomies of scale exist. This implies that about 70% of the dwellings face diseconomies of scale. Unfortunately, it is to be expected that economies of scale in this context relate to the organisational structure of the corporation and has less to do with the occupancy rate of the housing stock.

Wolters and Verhage [22] note that the estimation of efficiency of housing corporations is being hindered by heterogeneity, for example, because of differences in their working area or the composition of the housing stock.

Another closely related research line is about energy use in buildings. This type of research relates to the link between types of buildings, energy use and emissions. An interesting example can be found in Khoshbakht et al. [23]. They performed an analysis of higher education buildings of 80 university campus buildings in Australia. Energy use, energy use intensity, related space types and occupancy conditions were analysed using stochastic frontier analysis (SFA).

## 3. Methodology

We analyse the cost structure, scale economies and relative efficiency of Dutch municipality school buildings by applying stochastic frontier analysis (SFA) to a cost model with a panel data technique. The stochastic frontier approach goes back to the seminal work of Aigner, Lovell and Schmidt [24,25] and Meeusen and Van den Broeck [26], who

proposed a method that measures the distance of a specific firm to the firm that is operating at full-production maximisation or cost minimisation. The essence of this method is that this distance is a positive number that can be represented by a stochastic variable with a one-sided distribution. Since then, a pile of research emerged on this subject. For interesting oversights, see for example [27,28]. In this paper, we apply a cost model. A cost model relates minimum cost ($c$) to services delivery ($y$), input prices ($w$) and contextual variables ($z$). The basic model can be represented as follows:

$$\ln(c) = g(\ln(y), \ln(w), \ln(z), t) + v + u \tag{1}$$

where $g(\cdot)$ is a parametric specification of a cost function, $v$ reflects random errors and $u$ is a one-sided distributed efficiency component. Since both $v$ and $u$ are not observable, we use the panel data structure to disentangle efficiency from random errors. Several panel data estimation techniques can be applied [29,30]. Since we have a set of cross sections at our disposals, each containing a substantial number of observations, a random effects approach seems to be most appropriate. Due to the incidental parameter problem, a fixed effects approach would be less appropriate.

For the functional specification of $g(\cdot)$, we apply a translog function, which is a second-order Taylor approximation of a general function and is popular in empirical work due to being relatively flexible [31], as follows:

$$\ln(c) = a_0 + \sum_m b_m \ln(y_m) + \frac{1}{2} \sum_m \sum_{m'} b_{mm'} \ln(y_m) \ln(y_{m'}) + \sum_k d_k \ln(z_k) + \sum_k \sum_{k'} d_{kk'} \ln(z_k) \ln(z_{k'}) + h_1 t + h_{11} t^2 + \sum_m g_m \ln(y_m) t + \epsilon \tag{2}$$

where
$c$ = nominal cost deflated by a capital price index,
$y_m$ = production of services $m$,
$z_k$ = contextual variable $k$,
$t$ = trend,
$\epsilon$ = random error,
$a_0$, $b_m$, $b_{mm'}$, $d_k$, $d_{kk'}$, $h_1$, $h_{11}$, $g_m$ parameters to be estimated.

The cost function includes a time trend and a time squared variable allowing technical change to vary over time to a certain extent. Since we are analysing data on a time span of twelve years, this makes sense to do so. Furthermore, we assume an output biased technical change, i.e., that scale economies may change as a result of technological shifts. The data section will elaborate upon the choice of variables.

Further, we know that the cost of buildings, in particular the cost components related to construction, depend on, for instance, soil conditions and the easiness of access to the building site. These contextual variables are reflected by the z-variables.

From the estimated parameters of the cost function, we may derive several interesting economic outcomes.

### 3.1. Efficiency Scores

Efficiency estimates are obtained using the estimated errors, thereby shifting the errors by the maximum error in each specific year.

$$u_{it} = \epsilon_{it} - \min_i \epsilon_{it} \tag{3}$$

The efficiency score then equals the following:

$$Eff_{it} = \exp(-u_{it}) \tag{4}$$

### 3.2. Economies of Scale

Economies of scale are defined by the curvature of the estimated cost frontier with respect to $y$. Under (dis)economies of scale, an expanding output decreases (increases) the

average cost. The cost elasticity of output along a ray from the expansion path is equal to the following:

$$\eta = \sum_m \frac{\partial \ln(c)}{\partial \ln(y_m)} \tag{5}$$

By definition, it then holds that (dis)economies of scale exist for $\eta < 1$ ($\eta > 1$).

### 3.3. Technical Change

Technical change is defined as the relative change in costs in the course of time, wherein time here is seen as a reflection of the state of technology. This yields the following:

$$tc = \frac{\partial \ln(c)}{\partial t} \tag{6}$$

Equations (5) and (6) can be evaluated at different points in the output space and context space.

### 4. Choice of Variables and Data

In our cost model we distinguish the following three types of variables: cost, outputs, input prices and context variables. Costs are represented by capital cost, including depreciation, interest and some additional control cost. To derive the actual volume of capital input capital cost is deflated by a price index based on the depreciation rate, the interest rate and a price index for investment in fixed assets. Some measurement errors may occur due to quality differences in buildings. Unfortunately, we do not have any data at our disposal to control for these quality differences. Since the primary goal of school buildings is to provide shelter to the pupils of a school, the number of pupils obviously is a good candidate for an output measure. Contextual variables are variables that are out of the control of the municipalities but may affect the cost substantially. In the case of construction, population density and soil factor are important contextual factors. It is well-known that the construction of buildings in urban areas is more complex than in rural areas. In large parts of The Netherlands, construction work has to deal with soft soil requiring the use of piles and complex foundations.

The data come from the municipal accounts (Iv3), as collected and published by Statistics Netherlands. The municipal accounts include information on the school buildings costs of municipalities over the time period 2005–2016. These data are extensively checked and corrected. The municipalities with remarkably high fluctuations and unrealistic costs have been systematically removed from the analysis. This indicates that some municipalities are using a different accounting method for dealing with capital cost, the so-called investment a fonds perdu method, instead of the common method of annual depreciation. The final analysis includes 4929 observations. Since some of the observations had to be removed, the panel is unbalanced. Table 1 summarises the included variables and their descriptives. In order to obtain an impression of the number and size of Dutch municipalities, we also included a frequency table of the size of the municipalities measured by the number of inhabitants in 2016.

From Table 1 we notice that there is a substantial variation in variables between the almost 5000 observations. Capital costs vary between €27,000 and €86 million. The smallest municipality only has to take care of 58 pupils, whereas the biggest municipality has to shelter more than 66,000 pupils. There is also a large variation in population density. This large variation in outputs and costs once more underlines the importance the central issue of this paper regarding economies of scale. If scale economies prevail, then large savings by exploiting these scale economies may be expected.

From the frequency tabulation in Table 1, we notice that there are about 400 municipalities. About 6% of these municipalities could be qualified as small (<10,000 inhabitants), 50% as moderate–small (10,000–30,000 inhabitants), one third as moderate–big (30,000–100,000 inhabitants) and 8% as big (>100,000 inhabitants).

**Table 1.** Descriptive variables.

| Variable | N | Mean | Std. Dev. | Min | Max |
|---|---|---|---|---|---|
| 2005–2016 | | | | | |
| Capital cost | 4929 | 1909.7 | 4456.7 | 27.4 | 86,203.3 |
| Price of capital | 4929 | 1.135 | 0.173 | 0.845 | 1.495 |
| Number of pupils | 4929 | 3800 | 5686 | 58 | 66,579 |
| Population density | 4929 | 978 | 724 | 111 | 6094 |
| Soil factor | 4929 | 1.094 | 0.149 | 1.000 | 1.860 |
| 2016 | | | | | |
| Size | Freq. | Percent | Cum. | | |
| <10,000 | 23 | 5.96 | 5.96 | | |
| 10,000–30,000 | 202 | 52.33 | 58.29 | | |
| 30,000–100,000 | 131 | 33.94 | 92.23 | | |
| >100,000 | 30 | 7.77 | 100 | | |

## 5. Results

This section presents the results obtained from estimating the cost function (Equation (2)). The model estimates are presented in Table 2. The explanatory variables are standardised on the mean and taken in logarithms.

**Table 2.** Estimates based on random effects method.

| | Coef. | Std. Err. | Z-Score | Sign. |
|---|---|---|---|---|
| Number of pupils | 1.069 | 0.020 | 52.360 | 0.000 |
| Number of pupils × number of pupils | 0.121 | 0.021 | 5.770 | 0.000 |
| Soil density | 0.517 | 0.197 | 2.630 | 0.009 |
| Soil density × soil density | −3.517 | 1.418 | −2.480 | 0.013 |
| Number of pupils × soil density | 0.096 | 0.133 | 0.730 | 0.468 |
| Time | 0.060 | 0.006 | 10.820 | 0.000 |
| Time × time | −0.003 | 0.000 | −6.670 | 0.000 |
| Time × number of pupils | −0.003 | 0.002 | −1.790 | 0.074 |
| Time × soil density | −0.018 | 0.013 | −1.400 | 0.162 |
| Constant | −0.527 | 0.024 | −22.070 | 0.000 |
| Explained variance | 0.884 | | | |

From Table 2, we determine that the estimates make sense. Almost all the estimated parameters are significant at the 5% level. The only variable that does not contribute to the explanation of cost from a statistical point is population density and is, therefore, left out in the final model. The estimated model shows a good fit. When we include the random effects, the explained variance of the estimated cost equals 0.88. We now further evaluate the estimates by inspecting the various economic indicators that can be derived from them.

### 5.1. Economies of Scale

To analyse the influence of scale on building input, individual cost elasticities are derived according to Equation (4) and applied to all the municipalities in 2016. The outcomes are illustrated in Figure 1.

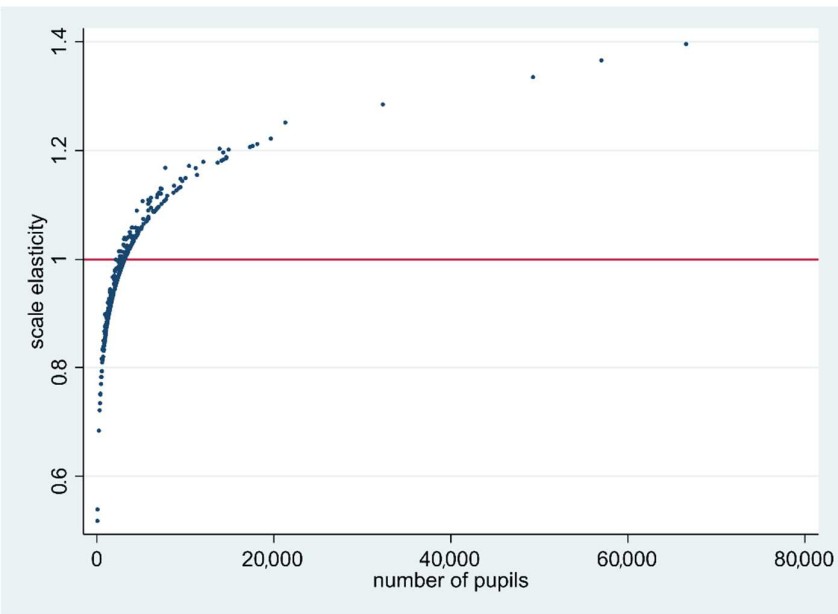

**Figure 1.** Elasticity of scale by number of pupils, 2016.

In Figure 1, it is demonstrated that a large part of predicted cost elasticities of scale equal about one, indicating that many municipalities face constant economies of scale. Applying a 95% two-sided statistical test, it is shown that about 41% of the municipalities face economies of scale, 39% constant economies of scale and 20% diseconomies of scale. According to Table 1, this implies that most of the municipalities with less than 30,000 inhabitants face economies of scale and at least all the municipalities greater than 100,000 inhabitants face diseconomies of scale. This implies that about 60% of the municipalities face some kind of scale inefficiencies. Therefore, there is room for improvement.

The existence of economies of scale in school buildings for small municipalities is due to the obligation to exploit small school buildings. Obviously, there is a direct correlation between small municipalities and low numbers of pupils. In The Netherlands, there is a strong preference for the proximity of schools in order to bring your children to school "just around the corner of the street" as well an unfamiliarity with the concept of school busses. Schools are also regarded as a part of the social coherence in small communities. In small school buildings, a relatively large "overhead" in space (hall, corridors, offices, utility rooms, etc.) exists. A low number of pupils per school also implies a low number of pupils per classroom and an inefficient usage of space. Another important explanation for the established scale effects is presumably the occupancy rate. Smaller municipalities or school boards have more difficulty absorbing fluctuations in the demand for school buildings.

*5.2. Efficiency Scores*

Figure 2 reflects the distribution of the estimated efficiency scores in 2016 by kernel density.

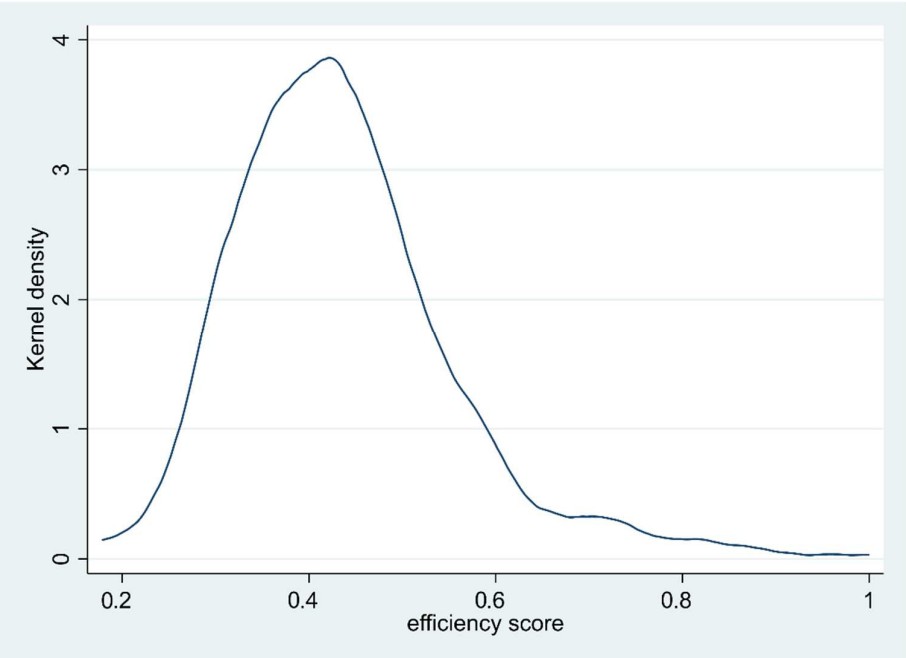

**Figure 2.** Kernel density of efficiency scores, 2016.

From Figure 2, it can be seen that the median efficiency score is a little above 0.4, indicating that large inefficiencies exist, and substantial improvements can be realised. In general, inefficiencies in capital utilisation are not very uncommon, since capital goods are fixed for a long time and cannot easily be adapted to changing needs. In particular, school buildings cannot easily be sold or utilised for purposes other than education due to their specific architecture. A substantial part of the inefficiencies is also the result of the fact that each classroom has a fixed capacity and will seldom be used to its full capacity due to the varying number of pupils in each class. Classrooms are constructed to shelter about a maximum of 30 pupils, but in practice will frequently be used only by 20 pupils or even less. Only in coincidental cases will the space in a school be used to its full potential. The long, small tail at the right side of Figure 2 reveals this aspect. Another explanation may be due to some special characteristics of the Dutch education system. Due to constitutional rights, parents can found new schools as they wish (and fulfil some requirements with respect to the number of pupils, etc.) based on religious or educational grounds. The government is obliged to fund these new schools and municipalities are responsible for providing school buildings. Although school boards cannot claim to have a school building of their own, it is quite common to have some physical separation in school buildings for different schools. This leads to extra inefficiencies.

*5.3. Technical Change*

Productivity also changes due to technical change. Figure 3 shows the relationship between productivity and time, reflecting productivity change due to technical change. The outcomes are calculated for schools reflecting an average size and average soil density.

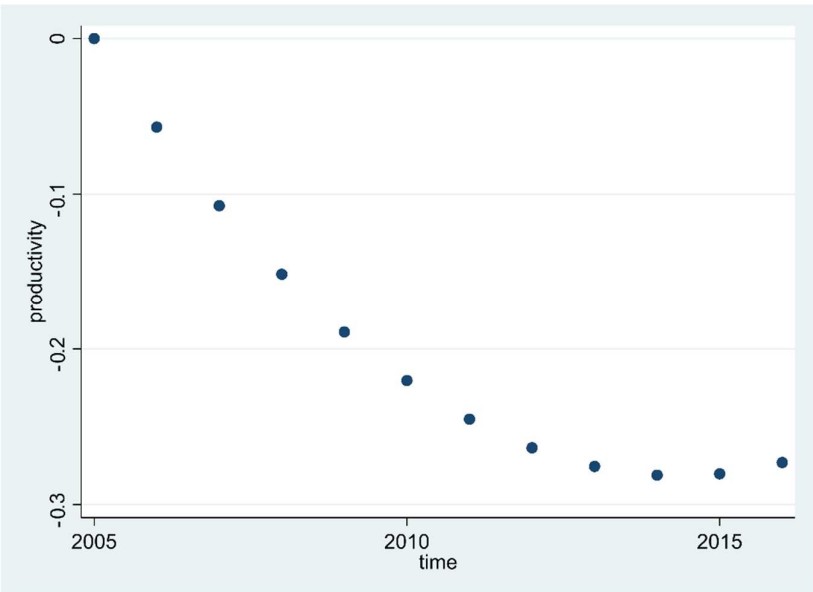

**Figure 3.** Productivity due to technical change, 2005–2016.

From Figure 3, we note that technical change is negative, indicating that over time more and more space has not been used efficiently. It also shows that this negative trend slows down over time and came to an end in 2015. Between 2005 and 2015, a loss of 28% in the productivity of school building productivity occurs. From this, we may conclude that incentives for productive building planning and management have been lacking.

## 6. Discussion and Conclusions

In this paper, we analysed the sustainability of local governments in providing school building capacity by applying a cost function model and the use of municipality data on the costs of school buildings and school enrolment.

From the results, we may conclude that school building planning and management in The Netherlands is far from productive. Small municipalities with low volumes of school building capacity and very large municipalities with high volumes show substantial scale inefficiencies. It is, therefore, recommended for small municipalities to collaborate on school building provision. From research regarding other public services, we know that collaboration may be very profitable [19,32,33]. In these cases, scale economies are materialised by collaboration instead of merging.

Even worse outcomes are related to technical inefficiencies. These inefficiencies amount to about 60%, indicating that substantial gains can be achieved. It must be noted that these outcomes may partly come from the special features of the educational process. Classrooms may not be used to their full extent due to varying class sizes. The number of classes may substantially vary over different school years and the use of the buildings is only limited to teaching hours during the day. Nevertheless, it must be emphasised that some municipalities do perform much better than other municipalities. These municipalities may be regarded as role models and other municipalities may learn from these role models.

Striking is the result that, in the period 2005–2015, municipalities experienced a decreasing performance each year. Finally, in 2015, we notice that productivity stabilised. There is obviously a lack of incentives for innovative behaviour regarding building planning and management. The fact that the negative trend is flatting out over the research period may be due to Bowen's Law. A free interpretation of Bowen's Law is that productivity change is inversely related with the available funding. In 2015, major reforms took place in the funding of local government associated with large budget cuts. These budget

cuts were already announced at a very early stage. Probably, the municipalities anticipated these budgets cuts in the years before 2015.

Note that aside from the abundant use of materials during construction, the unproductive provision of building capacities also implies an abundant energy consumption and maintenance during its operational lifetime. From a sustainability perspective, a substantial reduction in raw materials and the emission of carbon dioxide during the construction process and period of operation can be achieved. In this paper, we only focused on school buildings, but we may expect similar results in the provision of government buildings, police buildings, prisons and courts. These would be fruitful avenues for future research.

**Funding:** This research received no external funding.

**Institutional Review Board Statement:** Not applicable.

**Informed Consent Statement:** Not applicable.

**Data Availability Statement:** The data come from the municipal accounts (Iv3), as collected and published by Statistics Netherlands.

**Conflicts of Interest:** The author declares no conflict of interest.

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
