# Peer review of "Sustainable Provision of School Buildings in The Netherlands: An Empirical Productivity Analysis of Local Government School Building Operations"

_sustainability, doi:10.3390/su13169138_

Round 1

Reviewer 1 Report

The research was carried out at a fairly high level. However, the article could be improved. The review of literature on the topic (pages 96-100) is too short. It needs to be expanded. References that relate to public services have been studied. It should also describe the analysis of energy efficiency in the construction and operation of buildings, the problems of green building, etc. In the author's conclusions, one should correlate them not only with the activities of municipalities, but also show their place in the broader context of green building.

Reviewer 2 Report

1. Overall

This paper deals with the inefficiency of ‘school building’ in the Netherland. I think that the author gives practically important implications via the research. 

However, I do not find an academic significance or contribution as an academic paper that well deserves to be published in this journal. 

Author(s) need to clarify why the case, school building, is chosen for the research. The only answer is found in the manuscript, that is “no literature focusing on efficiency, scale economies or technical in the provision of school buildings has been produced.” But, it is not enough to explain why this case uniquely important compared to previous literature. If the author(s) found any different research outcome that had not been identified in prior studies, this research is good for an academic paper. I suggest that the author(s) add discussions pertaining to comparisons between this paper’s analysis and previous literature’s one. 

In addition, this paper is not kind for international readers who do not know Netherland well. How many municipalities are there? Do local governments have similar political systems? What are the results of decentralization? Do local governments have fiscal autonomy or discretion against the central government in the Netherland?  

2. Literature review

The literature review sections are not enough to cover the research topic. Only 5 papers were reviewed. More detailed reviews regarding School building, SFA, the scale of economy, and efficiency need to be supplemented. 

3. Analysis

The author(s) used five variables. Please show the theoretical background of why only the variables are suitable for the cost function. 

Table 2 shows regression results. I think that the table should show information about the goodness of fit. 

In Table 1, Capital cost and Number of pupils look abnormal, because their Mean values are smaller than Std.Dev. Please check if there are outliers in the two variables. 

Please add a Table regarding lines 228-233. 

Minor comments

Typo: 118, 128

Citation style: 128, 289

Omitted citation: 289

Reviewer 3 Report

I consider that the topic is well founded and the subject is well argued. I think the period could be extended for a better accuracy of the results. I think also that the results can be reported to other studies.

Round 2

Reviewer 2 Report

The revised manuscript has addressed most of my comments. Thank author(s) for your efforts.

One thing that the author refused is about interpretations of Figure 1. I thought this manuscript would be better if the author provides more detailed information. So, I suggested a table for example. Because the author classified municipalities into three groups regarding economies of scale: 41%, 39%, and 20%. Though it is redundant to add all localities in the article, at least a summarized table can be shown.

This comment is not the critical issue. So, it depends on the author’s decision. However, I strongly recommend it.

Before proof, please check the trivial errors below:

Line 141-142(revised manuscript): citation style error?

“An interesting example can be found in Khoshbakht et al. (2018).[23].”

131-132 (revised manuscript)

“He shows that the range for the optimal scale varies between 501 en 1,000 dwellings.”

Line 142-143(revised manuscript): performedan?

“They performedan analysis of higher education buildings of 80 university campus buildings in Australia.”
